# A Graph Convolutional Network-Based Deep Reinforcement Learning Approach for Resource Allocation in a Cognitive Radio Network

**DOI:** 10.3390/s20185216

**Published:** 2020-09-13

**Authors:** Di Zhao, Hao Qin, Bin Song, Beichen Han, Xiaojiang Du, Mohsen Guizani

**Affiliations:** 1The State Key Laboratory of Integrated Services Networks, Xidian University, Xi’an 710071, China; dizhao1002@gmail.com (D.Z.); hqin@mail.xidian.edu.cn (H.Q.); beichen.yt@gmail.com (B.H.); 2Department of Computer and Information Sciences, Temple University, Philadelphia, PA 19122, USA; dxj@ieee.org; 3Department of Computer Science and Engineering, Qatar University, Doha 2713, Qatar; mguizani@ieee.org

**Keywords:** cognitive radio, interference mitigation, resource allocation, dynamic graph, graph convolutional network, deep reinforcement learning, end-to-end learning model

## Abstract

Cognitive radio (CR) is a critical technique to solve the conflict between the explosive growth of traffic and severe spectrum scarcity. Reasonable radio resource allocation with CR can effectively achieve spectrum sharing and co-channel interference (CCI) mitigation. In this paper, we propose a joint channel selection and power adaptation scheme for the underlay cognitive radio network (CRN), maximizing the data rate of all secondary users (SUs) while guaranteeing the quality of service (QoS) of primary users (PUs). To exploit the underlying topology of CRNs, we model the communication network as dynamic graphs, and the random walk is used to imitate the users’ movements. Considering the lack of accurate channel state information (CSI), we use the user distance distribution contained in the graph to estimate CSI. Moreover, the graph convolutional network (GCN) is employed to extract the crucial interference features. Further, an end-to-end learning model is designed to implement the following resource allocation task to avoid the split with mismatched features and tasks. Finally, the deep reinforcement learning (DRL) framework is adopted for model learning, to explore the optimal resource allocation strategy. The simulation results verify the feasibility and convergence of the proposed scheme, and prove that its performance is significantly improved.

## 1. Introduction

With the deployment of the fifth-generation (5G) mobile communication system, users are provided with better quality of service (QoS) and quality of experience (QoE), with extremely high data rates and diversified service provisioning [1]. However, there are still many challenges in 5G wireless networks, such as the explosive growth of traffic and severe spectrum scarcity [2]. The conflict between the growing demand for wireless applications and the inefficient utilization of available radio spectrum resources can be resolved by a cognitive radio (CR). As a means to boost the performance of 5G wireless communication systems, CR has successfully attracted the attention of industry and academia.

The cognitive radio network (CRN) is an intelligent wireless communication system, which uses its cognitive ability to adjust its parameters according to the radio environment to dynamically access the available spectrum resources [3]. Apart from spectrum sensing and access, CR is also an intelligent technology with the capabilities of analysis and decision-making, for spectrum management and interference mitigation [4,5]. From this perspective, CR is considered as a novel radio spectrum resource allocation paradigm, in which unlicensed secondary users (SUs) opportunistically access the unused spectrum of licensed primary users (PUs), without interrupting the operation of the PUs [6]. Generally, there are three access paradigms for CRNs, including underlay, overlay, and hybrid. In the underlay mode, the concurrent transmission is allowed if the interference caused by SUs at PU receivers is under a predefined threshold known as the interference temperature [7]. In the overlay mode, SUs are only allowed to access the spectrum not used by PUs. The hybrid mode is a mixture of underlay and overlay. Moreover, the underlay paradigm is more efficient than the overlay paradigm in terms of the spectrum utilization, and is easier to implement than the hybrid paradigm [8]. Therefore, the underlay CRN will be adopted in this paper.

It is shown in a lot of work and research that the spectrum utilization can be significantly improved through reasonable resource allocation in the underlay CRN. However, due to the co-existence of primary base stations (PBSs) and secondary base stations (SBSs), the problem of co-channel interference (CCI) occurs, which poses a critical challenge for resource allocation in the underlay CRN. The severe CCI is caused by the characteristic that PUs and SUs share the same subchannel [9]. There are three sources of CCI, including interference from PU to SU, interference from SU to PU, and interference among SUs. Regarding the first type of interference, a reasonable threshold is commonly set to guarantee the QoS of PUs. For others, orthogonal transmission, power control, and interference constraints are usually used for interference elimination [10]. In this paper, we expect to achieve CCI mitigation through optimal resource allocation under interference constraints, including channel selection and power adaptation.

Currently, optimization theory and heuristic search are two prevalent tools of modeling and solving the resource allocation problems for CRNs. The work in [11] investigated energy-efficient resource allocation in orthogonal frequency division multiplexing (OFDM)-based CRNs, which is transformed to a non-linear fractional programming problem employing a time-sharing method, to obtain a near optimal solution by the standard optimization technique. The algorithm in [12] is designed to maximize the weighted sum-rate of orthogonally transmitting PUs, which is modeled as a non-convex optimization problem solved by the channel state information (CSI) of the primary and secondary networks and the Lagrange multipliers associated with the constraints. An energy-efficient optimization problem with the resource assignment and power allocation for the OFDMA-based H-CRNs is depicted as a non-convex objective function in [13], and closed form expressions for this problem are derived by the Lagrange dual decomposition method. In addition, the authors in [14] proposed the solution of resource allocation for CRN using the modified ant colony algorithm, which is a metaheuristic approximation inspired from the behavior of the colony of ants in foraging to the select channel. In [15], a dynamic media access control (MAC) frame configuration and optimal resource allocation problem for multi-channel and ad hoc CRN is presented and optimized by the particle swarm optimization (PSO) algorithm.

Even so, there are still many challenges existing in the resource allocation scheme for CRNs, which is generally NP-hard and near-optimal [16]. In real-time operation, the flaws of global constrained optimization, high computation time, and complexity will be highlighted.

To reduce the emergence of the above defects, CR users are expected to have learning capability to determine the optimal strategies for the CRNs. Fortunately, artificial intelligence (AI) technology has opened up a new world [17]. Machine learning (ML), especially reinforcement learning (RL), is envisioned as a potential solution that can be integrated with CR to obtain the optimal resource management strategy [18,19]. The authors in [20] discussed a novel approximated online Q-learning scheme for power allocation, in which cognitive users learn with conjecture features to select the most appropriate power level. The work in [21] proposes an asynchronous advantage actor critic (A3C)-based power control of SUs, and SUs learn power control scheme simultaneously on different CPU threads to reduce the interdependence of the neural network gradient update. Furthermore, it is known that simple individuals can attain significant abilities by swarm intelligence. Hence, RL technologies in multi-agent environments and distributed networks have become more and more popular. A multi-agent model-free RL scheme for resource allocation is presented in [22], which mitigates interference and eliminates the need of network model. This scheme is implemented in a decentralized cooperative manner with CRs acting as a multi-agent, forming a stochastic dynamic team to obtain the optimal strategy.

Although, the above work demonstrates that RL enables PUs and SUs of CRNs to intelligently occupy resources to improve spectrum utilization and energy efficiency. However, the fact is that the performance of these methods will degrade dramatically if the scale of the wireless communication network becomes larger [23]. The internal cause of this phenomenon is that RL algorithms usually define the state of the communication network as Euclidean data, including the CSI matrix and the users’ requests matrix, which fail to exploit the underlying topology of wireless networks. To make full use of topology information for effective learning, related work has been studied in depth. In [24], the authors explore the use of spatial convolution for scheduling under the sum-rate maximization criterion, while utilizing only location information. The work in [25] proposes a novel graph embedding-based method for link scheduling in D2D networks and develops a K-nearest neighbor graph representation method to reduce the computational complexity. Even though these methods are scalable to large-size wireless communication networks, the process of feature extraction and resource allocation are separated. It cannot be guaranteed that the extracted features will be most efficient for resource allocation tasks. In our work, an end-to-end learning model, namely the graph convolutional network (GCN), is adopted to explore the performance of resource management in the underlay CRN.

The main contributions of this paper are summarized as follows:We propose a method of constructing the topology of the underlay CRN based on a dynamic graph. The dynamics of the communication graph is mainly reflected in two aspects: One is to adopt the random walk model to simulate the users’ movements, which indicates the dynamics of the position of vertices; the other is the dynamics of the topology caused by the different resource occupation results. Moreover, a novel mapping method is also presented. We regard the signal links as vertices and interference links as edges. This simplifies the complexity of the graph, and is more suitable for extracting the desired interference information.Considering that it is difficult to obtain accurate CSI, we suggest an RL algorithm that utilizes graph data as state inputs. To make the state conditions looser, we model the path loss based on the user distance information inherent in the graph. Hence, the state can be defined by the user distance distribution and resource occupation, which substitute CSI. Additionally, the actions are two-objective, including channel selection and power adaptation, to achieve spectrum sharing and interference mitigation.We explore the performance of the resource allocation strategies with the “GCN+DRL” framework. Here, we design an end-to-end model by stacking the graph convolutional layers, to learn the structural information and attribute the information of the CRN communication graph. In this design, the convolutional layers are mainly used to extract interference features, and the fully connected layers are responsible for allocating the channel and power. The end-to-end learning model can automatically extract effective features, avoiding the mismatch between features and tasks. In this way, the reward of RL can simultaneously guide the learning process of feature representation and resource assignment.

The rest of this paper is organized as follows. Section 2 provides the system model and a detailed description of the optimization problem. Section 3 develops the resource allocation algorithm based on the DRL framework with GCN. Simulation results and analysis are discussed in Section 4. The conclusion is summarized in Section 5.

## 2. System Model and Problem Formulation

In this section, we first provide a detailed description of a CRN with the dynamic graph structure. From the perspective of graph, we then analyze the CCI existing in CRNs in depth. Finally, the formulation of the problem of spectrum-efficiency resource allocation is proposed to achieve spectrum sharing and interference mitigation.

### 2.1. System Model

As illustrated in Figure 1, the system model considered in our work is an underlay CRN, where the CR networks are underlaid with the coverage of the PU network. For simplicity, we assume that there is only one PU network with multiple PUs. We denote the set of PBSs as B={1,2,…,B} and SBSs as S={1,2,…,S}. The set of PUs is denoted as U={1,2,…U}. The set of SUs within each SBS coverage is defined as V={1,2,…,V}. All the underlay CR networks share the same radio resources with the PU network. Let N={1,2,…,N} denote the set of orthogonal resources with a total bandwidth W. We assume that only one PU is served on each resource block (RB) to avoid co-layer interference within the PU network. Meanwhile, multiple SUs compete to reuse the same RB to improve spectrum efficiency (SE). This multiplexing not only causes the co-layer interference within the CR networks but also causes severe cross-layer interference to the SUs and PUs, as shown in Figure 1. The main objective is to find the optimal resource allocation strategy for SUs, maximizing the data rate of the CR networks with the constraint that the interference caused to PUs is below a certain threshold.

#### 2.1.1. Path Loss Model

Considering the effects of multipath fading and shadow fading, we use the path loss model in [26]. The channel gain between BS and users can be expressed as:(1)$$h=\frac{10^{-\frac{K}{10}}}{{\left(4\pi f_{n}\zeta \right)}^{2}{\left(d\right)}^{\alpha }},$$
where K is a random variable representing the shadowing effect, and it is generally a Gaussian random variable with mean 0 and variance $\sigma^{2}$. $f_{n}$ represents the center frequency of channel n, and ζ is a correction parameter of the channel model. Besides, α is the path loss exponent, and d indicates the distance between users and the associated BSs. Substituting the distances mentioned above as d into Equation (1), the channel gains of the corresponding channel will be known. Hence, the channel gains of signal links from PBSs to PUs are expressed as $h_{u},\forall u\in U$, and the channel gains of signal links from SBSs to SUs are expressed as $h_{s,v},\forall s\in S,\forall v\in V$. Similarity, the channel gains of interference links from SBSs to PUs are represented as $h_{u}^{s,v},h_{s,v}^{u},\forall u\in U,\forall s\in S,\;\forall v\in V$, and the channel gains of interference links from unaffiliated SBSs to SUs are represented as $h_{\tilde{s},\tilde{v}}^{s,v},\forall s\in S,\forall v,\tilde{v}\in V,\;\forall \tilde{s}\in \left\{S\right\}\backslash s$.

#### 2.1.2. Dynamic Graph Construction Based on Users’ Mobility Model

In this work, we first model the CRN as a complete graph, as illustrated in Figure 2. To structure a complete graph of the underlay CRN, we need to extract the topology of the two-layer network. Here, we use the random walk model to simulate the movement of users. In this model, the direction of the motion is determined by an angle ϑ uniformly distributed between [0, 2π]. Besides, each user is assigned a random speed δ between $\left[0,\delta_{max}\right]$, and $\delta_{max}$ is the maximum speed of a user.

Based on this, we mark the real-time locations of the main components, including the BSs and users at each layer. We denote the locations of PUs as $\left(X_{u},Y_{u}\right),\forall u\in U$ and SUs as $\left(X_{v}^{s},Y_{v}^{s}\right),\forall s\in S,\forall v\in V$. The locations of PBSs and SBSs are denoted as $\left(X_{b}^{B},Y_{b}^{B}\right),\forall b\in B$ and $\left(X_{s}^{S},Y_{s}^{S}\right),\forall s\in S$, respectively. Then, we can easily obtain the distances of the users relative to the corresponding BS based on the positions marked above. The distances from PBSs to PUs are $d_{u},\forall u\in U$, and the distances from SBSs to SUs are $d_{sv},\forall s\in S,\forall v\in V$. Moreover, users located at the edge of a cell may receive interference signals, which are transmitted from the BSs in the neighbor cells. Let $d_{s,v}^{u},\;\forall s\in S,\forall v\in V,\forall u\in U$ denote the distances from SBSs to PUs, and let $d_{s,v}^{\tilde{s},\tilde{v}},\forall \;s\in S,\forall \tilde{s}\in \left\{S\right\}\backslash s,\forall v,\tilde{v}\in V$ denote the distances from unaffiliated SBSs to SUs. Thus, the elementary topology of the underlay CRN can be obtained in this way.

### 2.2. Problem Formulation

The resource allocation problem in the underlay CRN has an optimization goal with constraints, to maximize the data rate of the SUs while maintaining the SINR of the affected PUs. This goal depends on certain factors, including SINR of SUs and SINR of PUs. Based on the channel gains in the graph structure above, the SINR of SUs and PUs can be obtained by a certain amount of calculations. However, we need to determine the sources of interference that impact users’ signals before calculating the SINR. Therefore, the distribution of CCI that may exist in the underlay CRN is explored.

As shown in Figure 3, the CCI suffered by any SU may come from two aspects, including the cross-layer interference of PU and the co-layer interference of other SUs. We assume that the SUs are transmitted using adaptive modulation and coding (AMC), in which the modulation scheme and channel coding rate are adjusted according to the state of the transmission link. Under this condition, the SUs can infer the state of the primary link, and occupy the spectrum resource purchased from the PU network based on the channel quality condition. Thus, in the process of transmitting parameter setting, each SU needs to consider the interference constraint requirements from the PU and other SUs. Note that the initial topology state of the CRN is fully connected is assumed. That is, we suppose that any SU will be interfered by all PUs, as well as all SUs attached to other SBSs. In this way, the virtual interference links of the full graph interconnection are established. Additionally, the actual interference links will be established as resource allocation proceeds, and the real topology of the graph will vary with the different resource allocation results.

We assume that the interfering PU and SUs share the same RB n, which is used by the vth SU (covered by the sth SBS). The interference perceived by the vth SU (covered by the sth SBS) can be written as:(2)$$I_{s,v}\left[n\right]=\sum_{u=1}^{U}h_{u}^{s,v}\left[n\right]P_{u}^{s,v}\left[n\right]+\sum_{\tilde{s}\in \left\{S\right\}\backslash s}\sum_{\tilde{v}\in V}h_{\tilde{s},\tilde{v}}^{s,v}\left[n\right]P_{\tilde{s},\tilde{v}}^{s,v}\left[n\right],$$
where $P_{u}^{s,v},\forall u\in U,\forall s\in S,\forall v\in V$ is the transmission power of the interference link from PU to SU, and $P_{\tilde{s},\tilde{v}}^{s,v},\forall s\in S,\forall \tilde{s}\in \left\{S\right\}\backslash s,\forall v,\tilde{v}\in V$ is the transmission power of the interference link from SU of unaffiliated SBS to SU.

The received signals at the vth SU (covered by the sth SBS) contain four parts, which are the signal from the SBS, the interference signals from the PU and other SUs, and the channel noise. Consequently, the SINR for the vth SU (covered by the sth SBS) over RB n is given by:(3)$$\xi_{s,v}\left[n\right]=\frac{h_{s,v}\left[n\right]P_{s,v}\left[n\right]}{\sigma^{2}+I_{s,v}\left[n\right]}=\frac{h_{s,v}\left[n\right]P_{s,v}\left[n\right]}{\sigma^{2}+\sum_{u=1}^{U}h_{u}^{s,v}\left[n\right]P_{u}^{s,v}\left[n\right]+\sum_{\tilde{s}\in \left\{S\right\}\backslash s}\sum_{\tilde{v}\in V}h_{\tilde{s},\tilde{v}}^{s,v}\left[n\right]P_{\tilde{s},\tilde{v}}^{s,v}\left[n\right]},$$
where $P_{s,v},\forall s\in S,\forall v\in V$ is the transmission power of the signal link from SBS to SU, and $\sigma^{2}$ is the power of the additive white Gaussian noise (AWGN).

Considering that SUs similarly cause interference to the PU occupying the same RB n, we should maintain the interference below a tolerable threshold. The interference perceived by the uth PU over RB n can be expressed as:(4)$$I_{u}\left[n\right]=\sum_{s=1}^{S}\sum_{v=1}^{V}h_{s,v}^{u}\left[n\right]P_{s,v}^{u}\left[n\right],$$
where $P_{s,v}^{u},\forall u\in U,\forall s\in S,\forall v\in V$ is the transmission power of the interference link from SU to PU. Hence, the SINR for the uth PU over RB n is defined as:(5)$$\xi_{u}\left[n\right]=\frac{h_{u}\left[n\right]P_{u}\left[n\right]}{\sigma^{2}+\sum_{s=1}^{S}\sum_{v=1}^{V}h_{s,v}^{u}\left[n\right]P_{s,v}^{u}\left[n\right]},$$
where $P_{u},\forall u\in U$ is the transmission power of the signal link from PBS to PU. Given the SINR, the data rate of the vth SU (covered by the sth SBS) over RB n according to Shannon’s formula can be written as:(6)$$C_{s,v}\left[n\right]=x_{s,v}^{n}*\frac{W}{N}log_{2}\left(1+\xi_{s,v}\left[n\right]\right),$$
where $x_{s,v}^{n}$ is a binary indicator variable that denotes the assignment of RB n for the vth SU (covered by the sth SBS). $x_{s,v}^{n}=1$ represents that RB n is utilized by the vth SU (covered by the sth SBS), and $x_{s,v}^{n}=0$ otherwise. Here, we assume that any SU can simultaneously occupy more than one RB to maximize the data rate. Then, the total data rate of the vth SU (covered by the sth SBS) is given by:(7)$$C_{s,v}=\sum_{n=1}^{N}C_{s,v}\left[n\right]=\sum_{n=1}^{N}x_{s,v}^{n}*\frac{W}{N}log_{2}\left(1+\xi_{s,v}\left[n\right]\right).$$

Therefore, the achievable data rate of all CR networks can be obtained by:(8)$$C_{total}=\sum_{s=1}^{S}\sum_{v=1}^{V}C_{s,v}.$$

As mentioned earlier, our main objective is to maximize the data rate of the CR networks, while restricting the interference caused to PUs below a certain threshold. So, the spectrum-efficient resource allocation problem is formulated as an optimization problem as follows:(9)$$\begin{array}{c} \max\;C_{total}=\sum_{s=1}^{S}\sum_{v=1}^{V}C_{s,v}\\ s.\;t.\;C1:\;\sum_{v=1}^{V}x_{s,v}^{n}\leq 1,x_{s,v}^{n}\in \left\{0,1\right\},\forall n\in N,\forall s\in S\\ C2:\;\sum_{u=1}^{U}x_{u}^{n}\leq 1,x_{u}^{n}\in \left\{0,1\right\},\forall n\in N,\forall u\in U\\ C3:0\leq P_{u}\leq P_{b}^{\max},\forall u\in U,\forall b\in B\\ C4:\;0\leq P_{s,v}\leq P_{s}^{max},\forall s\in S,\forall v\in V\\ C5:\;0\leq P_{u}^{s,v},P_{s,v}^{u}\leq P_{s}^{max},\forall u\in U,\forall s\in S,\;\forall v\in V\\ C6:\;0\leq P_{\tilde{s},\tilde{v}}^{s,v}\leq P_{\tilde{s}}^{max},\forall s\in S,\forall \tilde{s}\in \left\{S\right\}\backslash s,\forall v,\tilde{v}\in V\\ C7:\;I_{u}\left[n\right]=\sum_{s=1}^{S}\sum_{v=1}^{V}h_{s,v}^{u}\left[n\right]P_{s,v}^{u}\left[n\right]\leq I_{u}^{th},\forall u\in U,\forall n\in N, \end{array}$$
where $x_{u}^{n}$ is also a binary indicator variable that denotes the assignment of RB n for the uth PU, and $P_{b}^{max}$ and $P_{s}^{max}$ are the maximum transmission powers of the PBS and SBS, respectively. Besides, $I_{u}^{th}$ represents the SINR threshold for the uth PU. The constraint in C1 indicates that RB n can only be occupied by one SU (covered by the sth SBS) at most, and other SUs under this SBS use orthogonal channels. The constraint in C2 means that the number of RBs selected by each PU should be at most one, to avoid CCI among PUs. Furthermore, the constraints C3, C4, C5, and C6 ensure that the power allocation of PUs and SUs do not exceed their respective maximum allowed transmission powers. Finally, the interference caused to PUs by SUs on certain RB is limited by a predefined threshold in the constraint C7.

## 3. Graph Convolutional Network-Based Deep Reinforcement Learning Approach for Resource Allocation in Cognitive Radio Network

In this section, the details of the spectrum-efficiency resource allocation algorithm for an underlay CRN is provided, which is a DRL approach with GCN. Firstly, we propose a brief introduction to RL and graph neural networks (GNNs), and GCN adopted in our paper is a type of GNN. Then, we illustrate how to define the critical RL elements in the resource allocation problem in an underlay CRN. Afterwards, we describe the procedure that the agent generates actions based on the GCN, including feature extraction and policy generation. Finally, the training process is presented.

### 3.1. Preliminaries

#### 3.1.1. Reinforcement Learning

RL is a learning process that guides the agent to take actions to maximize the long-term benefits based on a “reward” mechanism. The learning process can be modelled as a Markov decision process (MDP), which is defined as (O,A,R,P,γ). Given an observation o∈O, the agent will perform an action a∈A that produces a transition p∈P to a new observation $o^{\prime }\in O$, and will provide the agent with a reward r. The agent is designed to determine the rule for taking an action at a given observation, which is known as policy. The goal of the agent is to learn a policy π: O→A, to maximize the cumulative reward $R_{t}=\sum_{k=0}^{\infty }\gamma^{k}r_{t+k+1}$. Therefore, we have a relationship as follows:(10)$$\pi^{*}=argmax_{\pi }E_{\tau \sim \pi \left(\tau \right)}\left[R_{\tau }\right],$$
where γ is a discount factor that means the impact of the future returns on the current is somewhat weakened, and τ represents a trajectory obtained by interaction.

Policy gradient is a policy-based RL algorithm that optimizes by expressing the goal into a function of strategy parameters. Specifically, we parameterize the optimization objective and find the gradient of the strategy parameters θ. By updating in the direction of the gradient rise step by step, the strategy can be promoted to achieve the best. Based on this, the objective function of maximizing the expectation of the cumulative rewards is defined as:(11)$$J\left(\theta \right)=E_{\tau \sim \pi \left(\tau \right)}\left[R_{\tau }\right].$$

The gradient of the objective function is given by:(12)$$\nabla_{\theta }J\left(\theta \right)=\frac{1}{Z}\sum_{i=1}^{Z}[\sum_{t=0}^{T}\nabla_{\theta }log\pi_{\theta }(a_{i,t}|o_{i,t})\sum_{t=0}^{T}r\left(o_{i,t},a_{i,t}\right)],$$
where Z is the total number of episodes. To enhance the stability of the algorithm, the formula of the gradient can be further improved as follows:(13)$$\nabla_{\theta }J\left(\theta \right)=\frac{1}{Z}\sum_{i=1}^{Z}\sum_{t=0}^{T}[\nabla_{\theta }log\pi_{\theta }(a_{i,t}|o_{i,t})\left(\sum_{t^{\prime }=t}^{T}r\left(o_{i,t^{\prime }},a_{i,t^{\prime }}\right)-b_{i,t^{\prime }}\right)],$$
where $b_{i,t^{\prime }}$ is the baseline, to reduce the fluctuation of the algorithm without affecting the expected value.

#### 3.1.2. Graph Neural Networks

In general, a graph is represented as a set of vertices C={1,2,…,C} and edges ε={1,2,…,E}, which is denoted as G=(C, ε). The graph signal is used to describe a mapping C→R. It can be depicted as $f={\left[f_{1},f_{2},\ldots ,f_{C}\right]}^{T}$, where $f_{c}$ is the signal strength on vertex c. Besides, there is an inherent associative architecture among vertices, so the topology also needs to be studied. An edge connecting $C_{i}$ and $C_{j}$ is denoted as $e_{ij}$. Meanwhile, $C_{j}$ is considered to be a neighbor of $C_{i}$. We use the adjacency matrix to describe this association, which is written as:(14)$$A_{ij}=\{\begin{array}{ll} 1, & if\;e_{ij}\subseteq \varepsilon \;\\ 0, & else \end{array}.$$

Moreover, the number of edges with $C_{i}$ as the end vertex is called the degree of vertex $C_{i}$. The degree matrix is a diagonal matrix, and is expressed as:(15)$$D_{ii}=\sum_{j}A_{ij}.$$

Thus, the Laplacian matrix is defined as L=D−A. The Laplacian matrix is the core of exploring the properties of the graph structure. It is given by:(16)$$L_{ij}=\{\begin{array}{c} \left|num\left(C_{i}\right)\right|,\;if\;i=j\\ -1,\;if\;e_{ij}\subseteq \varepsilon \\ 0,\;otherwise \end{array}.$$

GNN is a connection model, which captures the dependencies in the graph through the message propagation among vertices. Consequently, GNN can be used to deal with learning problems with a graph structure or non-Euclidean data. The learning goal of GNN is to obtain the hidden state embedding ${𝒽}_{c},c\in C$ based on graph perception. Specifically, GNN iteratively updates the hidden state representation of each vertex by aggregating features from its edges and neighboring vertices. At time t+1, the hidden state embedding of vertex c is updated as follows:(17)$${𝒽}_{c}^{t+1}=f\left(\chi_{c},\chi_{c}^{con\left(c\right)},\chi_{c}^{nei\left(c\right)},{𝒽}_{nei\left(c\right)}^{t}\right),$$
where f(·) is the local transaction function, $\chi_{c}^{con\left(c\right)}$ is the features of the edges adjacent to vertex c, $\chi_{c}^{nei\left(c\right)}$ represents the features of the neighbor vertices of c, and ${𝒽}_{nei\left(c\right)}^{t}$ represents the hidden state embedding of the neighbor vertices at time t.

In addition, the design of the fitting function f(·) is crucial and leads to different types of GNN. The main popular GNN categories are GCN, graph attention networks (GAT), graph autoencoders (GAE), graph generative networks, and graph spatial-temporal networks. One can refer to [27,28,29] for more detailed information.

### 3.2. Definition of RL Elements of the CRN Environment

The DRL framework of the resource allocation in an underlay CRN is illustrated in Figure 4. In the following, we will describe each component of the RL framework in detail.
Agent: Based on the “GCN+DRL” framework, a central controller is treated as the agent. In other words, our work adopts a centralized RL algorithm. Interacting with the environment, the agent has the ability to learn and make decisions. The central controller is responsible for scheduling spectrum and power resource for PUs and SUs in the underlay CRN.State: The state in the RL framework represents the information that the agent can acquire from the environment. To design effective states for the resource allocation problem in a CRN environment, the important insight is that, the sum data rate of the CRNs is significantly influenced by the CCI. Further, the CCI, including co-layer interference among SUs and cross-layer interference between SUs and PUs, is the result of the user distance distribution and resource occupation.

In our work, the states mainly consist of the user distance distribution matrix D and resource occupation matrix X. Therefore, the system state is defined as:(18)O(t)={D(t),X(t)},
where D(t) is a symmetric matrix composed of 4 submatrices, and X(t) includes 2 submatrices. The specific compositions are shown as:(19)$$D\left(t\right)=\left[\begin{array}{cc} D_{1}\left(t\right) & D_{2}\left(t\right)\\ D_{3}\left(t\right) & D_{4}\left(t\right) \end{array}\right]=\left[\begin{array}{cc} d^{U\times U} & d^{U\times \left(S*V\right)}\\ d^{(S*V)\times U} & d^{\left(S*V\right)\times \left(S*V\right)} \end{array}\right],$$
and:(20)$$X\left(t\right)=\left[\begin{array}{cc} X_{1}\left(t\right) & X_{2} \end{array}\left(t\right)\right]=\left[\begin{array}{cc} x_{u}^{n}{}^{N\times U} & x_{s,v}^{n}{}^{N\times \left(S*V\right)} \end{array}\right],$$
and the state definition is detailed in Table A1 and Table A2 in Appendix A.
Action: An action is a valid resource allocation process to satisfy users’ request. At every single step, we define the actions as channel and power allocation of all PUs and SUs, which can be expressed as:
(21)$$A\left(t\right)=\left\{\left[\begin{array}{cc} A_{c}\left(t\right) & A_{𝓅}\left(t\right) \end{array}\right]\right\}=\left\{\left[\begin{array}{cc} a_{c}^{\left(U+S*V\right)\times N} & a_{𝓅}^{\left(U+S*V\right)\times M} \end{array}\right]\right\},$$
where $A_{c}\left(t\right)$ is the channel selection matrix and $A_{𝓅}\left(t\right)$ is the power selection matrix. The values of the actions will be determined by the interaction with the environment. A reasonable resource selection achieves spectrum sharing and interference mitigation, while satisfying the constraints of the optimization problem mentioned above. In the channel selection process, $x_{u}^{n}$ and $x_{s,v}^{n}$ represent the occupation of the nth RB by the uth PU and the vth SU, respectively. In the power selection process, M={1,2,…,M} is denoted as the set of the power levels, and, $y_{u}^{m}$ and $y_{s,v}^{m}$ represent whether the mth power level is chosen by the uth PU and the vth SU. The specific forms of these two matrices are as follows:(22)$$A_{c}\;=\;\left[\begin{array}{ccccccccc} {}[x_{1}^{1} & & \ldots & & x_{1}^{n} & & \ldots & & x_{1}^{N}]\\ & \vdots & & \vdots & \vdots & \vdots & & \vdots & \\ {}[x_{u}^{1} & & \ldots & & x_{u}^{n} & & \ldots & & x_{u}^{N}]\\ {}[x_{1,1}^{1} & & \ldots & & x_{1,1}^{n} & & \ldots & & x_{1,1}^{N}]\\ & \vdots & & \vdots & \vdots & \vdots & & \vdots & \\ {}[x_{1,v}^{1} & & \ldots & & x_{1,v}^{n} & & \ldots & & x_{1,v}^{N}]\\ & \vdots & & \vdots & \vdots & \vdots & & \vdots & \\ {}[x_{s,1}^{1} & & \ldots & & x_{s,1}^{n} & & \ldots & & x_{s,1}^{N}]\\ & \vdots & & \vdots & \vdots & \vdots & & \vdots & \\ {}[x_{s,v}^{1} & & \ldots & & x_{s,v}^{n} & & \ldots & & x_{s,v}^{N}] \end{array}\right]\;A_{𝓅}\;=\;[\begin{array}{ccccccccc} {}[y_{1}^{1} & & \ldots & & y_{1}^{m} & & \ldots & & y_{1}^{M}]\\ & \vdots & & \vdots & \vdots & \vdots & & \vdots & \\ {}[y_{u}^{1} & & \ldots & & y_{u}^{m} & & \ldots & & y_{u}^{M}]\\ {}[y_{1,1}^{1} & & \ldots & & y_{1,1}^{m} & & \ldots & & y_{1,1}^{M}]\\ & \vdots & & \vdots & \vdots & \vdots & & \vdots & \\ {}[y_{1,v}^{1} & & \ldots & & y_{1,v}^{m} & & \ldots & & y_{1,v}^{M}]\\ & \vdots & & \vdots & \vdots & \vdots & & \vdots & \\ {}[y_{s,1}^{1} & & \ldots & & y_{s,1}^{m} & & \ldots & & y_{s,1}^{M}]\\ & \vdots & & \vdots & \vdots & \vdots & & \vdots & \\ {}[y_{s,v}^{1} & & \ldots & & y_{s,v}^{m} & & \ldots & & y_{s,v}^{M}] \end{array}].$$Reward: Instead of following a predefined label, the learning agent optimizes the behavior of the algorithm by constantly receiving rewards from the external environment. The principle of the “reward” mechanism is to tell the agent how good the current action is doing relatively. That is to say, the reward function guides the optimizing direction of the algorithm. Hence, if we correlate the design of the reward function with the optimization goal, the performance of the system will be improved driven by the reward.

In the resource allocation problem for CRN, we define the reward as the total data rate of the CR networks, which can be written as:(23)$$r\left(t\right)=C_{total}=\sum_{s=1}^{S}\sum_{v=1}^{V}C_{s,v}.$$

Generally, an action that satisfies all users’ requests without violating constraints is considered to be good and encouraged, and the agent will receive a positive reward. This means that the probability of selecting the current action should be enforced. On the contrary, an action that violates constraints or causes severe CCI is treated as failed and prevented, and a negative reward will be fed back to the agent. This implies that the agent has more possibilities to search for other resource allocation decisions. Consistent with maximizing cumulative discount rewards, the overall optimization goal is achieved by constantly promoting the resource allocation policy.

### 3.3. Resource Allocation Algorithm Based on a Graph Convolutional Network

#### 3.3.1. State Mapping Method Based on a Dynamic Graph

Based on the dynamic topology of the underlay CRN, we also need to convert the state information into graph structure data, as illustrated in Figure 5. At each time, we have to capture the topological status of the CRN network in real time. The detailed method is that, at first, the signal links are regarded as the vertices, and the feature inputs of each vertex include the distance from the user to BS, the moving speed, and direction of the user; then, the interference links of the whole graph are derived based on the resource occupation, and the interference links act as edges and are characterized by the distance between interfering users. In particular, the states as feature inputs of the graph will indirectly act on the CSI, which affects the interference pattern and intensity of the entire CRN. The features of vertices have an impact on $h_{u}$ and $h_{s,v}$, and the connection relationships within the PU and CR networks are considered in the graph. Additionally, the features of edges can affect $h_{s,v}^{u}$, $h_{u}^{s,v}$, and $h_{\tilde{s},\tilde{v}}^{s,v}$, and the associations between cross layers are also included in the graph.

In this way, the state information is converted into graph signals, which can be used to capture the essential features with the assistance of GCN. In other work, the general way is to regard the BSs and users as vertices, while the signal and interference links are unified as edges. Compared with this, our proposed method distinguishes the representation of signal and interference links, which simplifies the complexity of the graph. Only extracting the interference links as edges can more intuitively capture the pattern of CCI in the CRN. This facilitates the analysis of interference strength to complete the subsequent resource allocation tasks, which is detailed in Section 3.3.2.

#### 3.3.2. End-to-End Learning Model Integrated Feature Extraction and Policy Generation

In the solution to the resource allocation and interference mitigation problem, the spatial features of the underlay CRN topology are critical. As far as we know, the convolutional neural network (CNN) can extract and combine the multi-scale local spatial features to build highly expressive representations. However, the limitation of CNN is that it can only manage regular Euclidean data, including image and text processing. As for the topology of the communication network, it is non-Euclidean data. In this case, the number of neighbors of the vertex is not fixed, and it is difficult to use a learnable convolution kernel with a fixed size to extract features. CNN is not applicable for dealing with the spatial features of the underlay CRN topology. Therefore, a spectral-based graph convolution model is adopted in our work. The essence of graph convolution is to find a learnable convolution kernel suitable for graphs.

The spectral-based method introduces filters to define graph convolution, which is inspired by the Fourier transform. The traditional Fourier transform converts a function in the time domain to the frequency domain. It is denoted as:(24)$$F\left(\omega \right)=F\left(f\left(t\right)\right)=\int f\left(t\right)e^{-i\omega t}dt,$$
which can be regarded as the integral of the time-domain signal f(t) and the eigenfunction of the Laplace operator $e^{-i\omega t}$. Here, if the graph Laplace operator is found, the graph Fourier transform can be defined as a discrete integral, which is given by:(25)$$F\left(\lambda_{l}\right)=\hat{f}\left(\lambda_{l}\right)=\sum_{c=1}^{C}f\left(c\right)u_{l}\left(c\right),$$
where $\lambda_{l}$ is the lth eigenvalue of the graph Laplace operator. f is a C-dimensional signal vector on the graph, and f(c) corresponds to each vertex. $u_{l}\left(c\right)$ represents the cth component of the lth eigenvector. Furthermore, we can derive the matrix form of the graph Fourier transform, which is expressed as:(26)$$\hat{f}=U^{T}f,$$
where U is an orthogonal basis formed by the eigenvectors. Note that the graph Laplace operator is actually the Laplace matrix mentioned above. Reversely, the traditional inverse Fourier transform is defined as:(27)$$F^{-1}\left[F\left(\omega \right)\right]=\frac{1}{2\pi }\int F\left(\omega \right)e^{i\omega t}d\omega ,$$
and migrated to the graph, the graph inverse Fourier transform is written as:(28)$$f\left(c\right)=\sum_{l=1}^{C}\hat{f}\left(\lambda_{l}\right)u_{l}\left(c\right),$$
which can be similarly expressed in matrix form as follows:(29)$$f=U^{T}\hat{f}.$$

The theory of spectral-based graph convolution is that, firstly, the representation of vertices is mapped to the frequency domain by the Fourier transform; secondly, the convolution in the time domain is realized by product in the frequency domain; finally, the product of the features is mapped back to the time domain by the inverse Fourier transform. Here, the convolution theorem is applied, which can be formulated as:(30)$$F\left(f*h\right)=\hat{f}\left(\omega \right)\cdot \hat{h}\left(\omega \right).$$

Thus, the principle of spectral-based graph convolution is given by:(31)$$f*h=F^{-1}\left(\hat{f}\left(\omega \right)\cdot \hat{h}\left(\omega \right)\right)=\frac{1}{2\pi }\int \hat{f}\left(\omega \right)\cdot \hat{h}\left(\omega \right)e^{i\omega t}d\omega ,$$
and the matrix form is written as:(32)$${\left(f*h\right)}_{G}=U\left(\left(U^{T}f\right)⊙\left(U^{T}h\right)\right),$$
where ⊙ is the element-wise Hadamard product. $U^{T}f$ and $U^{T}h$ represent the Fourier transform of the original feature of the graph and the convolution kernel, respectively. Since the convolution kernel is self-designed and self-learned, $h_{\theta }=U^{T}h$ can be converted into a diagonal matrix. The spectral-based graph convolution can be further expressed as:(33)$${\left(f*h\right)}_{G}=U\left(\begin{array}{cccc} \hat{h}\left(\lambda_{1}\right) & & & \\ & \hat{h}\left(\lambda_{2}\right) & & \\ & & \ddots & \\ & & & \hat{h}\left(\lambda_{C}\right) \end{array}\right)U^{T}f.$$

As we all know, convolution in deep learning is to design a kernel with trainable and shared parameters. It can be seen intuitively from the Equation (33) that the convolution kernel in the graph convolution is $h_{\theta }=diag\left(\hat{h}\left(\lambda_{l}\right)\right)$. So, the expression of the graph convolutional layer is [30]:(34)$$y_{output}=\sigma \left(Ug_{\theta }\left(\Lambda \right)U^{T}x\right),$$
where σ(·) is the activation function, and $g_{\theta }\left(\Lambda \right)$ is the convolution kernel. For better spatial localization and computational complexity, the kernel filter is designed as $g_{\theta }\left(\Lambda \right)=\sum_{k=0}^{K-1}\theta_{k}\Lambda^{k}$, and the output of the graph convolutional layer is illustrated as [31]:(35)$$y_{output}=\sigma \left(\sum_{k=0}^{K}\theta_{k}L^{k}x\right),$$
where the property of eigen decomposition is applied, and $U\Lambda^{k}U^{T}=L^{k}$. Specially, K is the receptive field of the convolution kernel, and a K-hot neighborhood is introduced. Additionally, K can be set to 1, that is, only the direct neighborhood is considered in each graph convolutional layer. In this way, the width is reduced, while the depth must be deepened. The method is to expand the receptive field by stacking multiple graph convolutional layers [32].

In our work, an end-to-end model integrating representation learning and task learning is built based on GCN. The network structure of the end-to-end learning model is shown in Figure 6. To extract features on the underlay CRN topology, we use 2 graph convolutional layers with an order index of K=1. If too few stacked layers are set, the vertex will lack adjacent feature information. This is because the vertex can only identify and aggregate few neighbors. Conversely, if too many stacked layers are set, almost all vertices will be judged and shared as neighbors after multi-hop propagation. Consequently, each vertex of the graph will present a highly similar representation, which is undesirable for the resource allocation task. Even, “over smoothing” may occur in the training process. The interference features can be effectively extracted by stacking two GCN layers. Moreover, we use three fully connected layers as the local output function. In the RL framework, the main contribution of the local output function is to generate the probability distribution of the actions. The fully connected layers gradually adapt to the channel and power allocation by adjusting parameters. To interpret the output as a probability distribution, we take the softmax layer as the output layer. The softmax function converts an arbitrary real vector into a vector within a range of (0, 1), which is used as a reference for selecting actions. In particular, the actions are two-objective in our learning model. Our solution is to share the GCN layers and the fully connected layers of this model, and use different softmax layers to achieve the two sub-goals of channel selection and power adaptation. Additionally, the total loss function of the entire model is set as the sum of the losses of the two subtasks. In this way, the weights can simultaneously learn the strategies of channel selection and power adaptation through back propagation. The weight sharing approach can avoid the complexity of designing two learning models.

In addition, the advantage of the end-to-end model based on GCN is that the feature vectors of the vertices will not be solidified due to concatenated, and the reward of the RL framework can still guide the representation learning of graph data. Consequently, the most effective spatial features for resource allocation task can be automatically extracted. At the same time, the parameters’ updating process of the GCN layers and task layers will be carried out simultaneously driven by the reward of the entire model. The representation learning and task learning are integrated into one model for end-to-end learning. This enhances the cohesion of the two learning stages, and shows better adaptation to practical problems.

#### 3.3.3. Learning Process based on the Policy Gradient Algorithm

In the framework of “GCN+DRL”, the agent is responsible for generating action strategies, according to the states defined in Section 3.2. Based on the above, it can be known that the state is graph data generated in real time. To utilize the input state effectively, the GCN layers with a set of trainable parameters are applied to extract and represent the features. Then, the probability distribution over actions is generated by an approximator based on neural networks. The end-to-end learning model is actually the agent, which needs to sufficiently experience various states and actions and iteratively optimizes the resource allocation policy. To improve the policy, a policy gradient algorithm is adopted to train the parameters of the learning model.
**Algorithm 1: Resource Allocation Algorithm Based GCN+DRL in the Underlay CRN****begin** **Initialization:**  Each user is dropped randomly with an arbitrary speed δ and direction ϑ of movement  The parameter of CRN system model is initialized, and CSI is set to a random value  All RBs are initialized to the idle state  The policy network parameter θ is initialized **Processing:**  **For**
i
**in *Z*, do**   Initialize the underlay CRN environment   **For**
t
**in**
T**, do**    Construct the graph G=(C, ε) of current CRN topology    Observe the state O(t) from the communication graph, including user distance distribution D(t) and resource occupation X(t)
    Select channel $A_{c}\left(t\right)$, according to the $\pi_{\theta }^{c}(a_{i,t}^{c}|o_{i,t})$
    Select power level $A_{𝓅}\left(t\right)$, according to the $\pi_{\theta }^{𝓅}(a_{i,t}^{𝓅}|o_{i,t})$
    Perform channel selection and power control, and obtain the reward $r\left(o_{i,t^{\prime }},a_{i,t^{\prime }}\right)$ according to the data rate of the CR networks    Check SINR to guarantee QoS of users according to constraints    Establish the actual interference links based on the resource allocation result   **End for**   Calculate the loss of channel selection and power adaptation, $L_{\theta }^{c}$ and $L_{\theta }^{𝓅}$
   Calculate the total loss $L_{total}$
   Update the network parameter θ with the gradient descent method  **End for****end**

In the CRN environment, we perform channel selection and power control by the policy gradient algorithm, as shown in Algorithm 1. Firstly, to initialize the CRN environment. More specifically, users are randomly placed with a given speed and direction, the initial CSI is set to a random value, and all RBs are reset to an idle state. According to the proposed method of constructing a graph, the positions of all components in the CRN are captured at each moment; meanwhile, all virtual interference links are established. The graph is mapped to state inputs O(t), which consist of the user distance distribution D(t) and resource occupation X(t), and represented as spatial features. The agent interacts with the CRN environment and performs actions. The action strategy is approximated by an end-to-end learning model. This learning model integrates two stages of feature extraction and strategy generation. Then, the agent combines the output of the network, which is a probability distribution over all possible actions of channel selection $A_{c}\left(t\right)$ and power adaptation $A_{𝓅}\left(t\right)$, to achieve the optimization goals. Afterwards, the agent performs actions, and then a new round of changes occurs in the graph of CRN based on the result of resource assignments. As the resource allocation proceeds, the virtual interference links are replaced by the actual interference links. The above steps are repeated. Specially, the optimal action is unknown, and the performance after execution is judged by the reward r(t). During the learning process, the agent continuously updates the policy driven by the cumulative reward function, until the optimal resource allocation policy is learned. We optimize the cross-entropy loss and backpropagate the gradients through the policy network. The loss function of channel selection and power adaptation are:(36)$$L_{\theta }^{c}=\frac{1}{Z}\sum_{i=1}^{Z}\sum_{t=0}^{T}[log\pi_{\theta }^{c}(a_{i,t}^{c}|o_{i,t})\left(\sum_{t^{\prime }=t}^{T}r\left(o_{i,t^{\prime }},a_{i,t^{\prime }}^{c}\right)-b_{i,t^{\prime }}\right)],$$
and:(37)$$L_{\theta }^{𝓅}=\frac{1}{Z}\sum_{i=1}^{Z}\sum_{t=0}^{T}[log\pi_{\theta }^{𝓅}(a_{i,t}^{𝓅}|o_{i,t})\left(\sum_{t^{\prime }=t}^{T}r\left(o_{i,t^{\prime }},a_{i,t^{\prime }}^{𝓅}\right)-b_{i,t^{\prime }}\right)].$$

Therefore, the total loss is given by:(38)$$L_{total}=L_{\theta }^{c}+L_{\theta }^{𝓅}.$$

## 4. Simulation and Evaluation

In this section, we present experiments to evaluate our proposed resource allocation algorithm, which was implemented by a GCN-based DRL framework. The experiments were conducted in an Ubuntu operating system (CPU Intel core i7-7700 3.6 Hz; memory 8 GB; GPU NVIDIA GeForce GTX 1070 Ti, which contains 2432 CUDA computing core units and 8 GB graphics memory). As illustrated in Figure 1, we consider a cell where the CR networks underlaid with the coverage of the PU network. The value of the path loss model is based on [26], and the setting of the interference temperature refers to the minimum SINR in the literature [33]. All parameters are summarized in Table 1.

Firstly, we compare our proposed algorithm (GCN+DRL) with the following approaches: 1. Random strategy based on our proposed network structure (random strategy); 2. policy gradient algorithm with fully connected layers (PG algorithm); and 3. the CNN-based DRL method (CNN+DRL). A comparison of different algorithms for resource allocation in the underlay CRN is illustrated in Table 2. Figure 7 shows the achievable data rate of different algorithms. The performance of GCN+DRL proposed in this paper is the best. In terms of convergence, the convergence time of the random strategy is relatively shorter. However, this solution is not the optimal resource allocation scheme, since the achievable data rate is stable at [2700, 2800] (kbps). The policy gradient algorithm stabilizes at about 40,000 iterations, but there are large fluctuations due to user mobility. Moreover, it can be concluded that the mere RL method cannot learn the optimal solution. Further, we explored the performance of the CNN+DRL scheme. It failed to converge, since CNN can only tackle Euclid data, not exploiting the underlying topology of wireless networks. In comparison, the performance of the GCN+DRL scheme is the best. The convergence time is 8800s, and the optimal resource allocation strategy can be learned in a relatively short time.

Figure 8a depicts the convergence performance of our proposed joint channel selection and power adaptation algorithm. First of all, we performed an experimental verification of the convergence performance in case of the fixed user distance distribution. We showed the expected rewards per training step with increasing training iterations. When the learning network first started training, the values of the expected rewards were relatively small, and the algorithm was in the exploration phase. As the number of training processes increases, the value of expected rewards gradually increases. This demonstrates that the learning agent is learning and analyzing the historical trajectories. The expected rewards stabilize after training 20,000 iterations, which means that our algorithm will automatically update its decision strategy and converge to the optimal. The figure shows that the GCN-based DRL scheme has good convergence in the resource allocation algorithm for the underlay CRN, and the convergence time is short. Additionally, Figure 8b illustrates the total loss during the training process. It can be seen that after 5000 iterations, the loss drops to the minimum. However, there are slight fluctuations in the training process, and we believe that this does not affect the performance of our algorithm.

As shown in Figure 9, we studied the convergence performance of the proposed algorithm at different learning rates. We compared the two sets of learning rate settings. The learning rates of the first group are 0.0003, 0.0005, 0.0007, and 0.00001, and the second group are 0.00001, 0.00003, 0.00005, and 0.00007, respectively. It can be seen that there is a same trend among the curves of different learning rates, but the convergence time is slightly different. As far as the trend, the expected rewards are low in the early stage because the agent is mainly responsible for exploration, and then all the curves gradually rise and stabilize. More intuitively, the expected value with a learning rate of 0.00001 is the largest, which is stabilized at around 3100 kbps. In terms of convergence time, the curve with a learning rate of 0.00001 converges around 20,000 iterations, but the number of iterations for the convergence of the curves with learning rates of 0.00005 and 0.00007 is relatively small, around 18,000 iterations. It can be concluded that a relatively large learning rate can accelerate the learning process. Nonetheless, it can be seen from Figure 9a that it will cause an “oscillation” phenomenon (e.g., lr = 0.00003) if the learning rate is set too large. It is also possible that the algorithm converges to the local optimal value (e.g., lr = 0.00005 and 0.00007). Conversely, setting the learning rate too small will result in slow convergence. From Figure 9b, the curve (lr = 0.00001) converges to an approximate optimal value at 15,000 iterations, but nearly 22,000 iterations are used on the curve (lr = 0.000003). Then, to converge to the optimal learning strategy, we are more inclined to sacrifice the convergence time and choose a relatively small learning rate. Based on the above-mentioned factors, when the learning rate is 0.00001, the convergence performance is the best. Hence, we adopt the learning rate of 0.00001 in the following simulations.

In Figure 10, we compare the expected rewards of users in four groups of different neuron numbers. We set the learning rate to 0.00001. The number of neurons in the first graph convolutional layer is 8 × 4, 8 × 32, 8 × 32, 8 × 64, respectively. Additionally, the number of neurons in the second graph convolutional layer is 4 × 2, 32 × 64, 32 × 128, 64 × 128, respectively. It is shown in the figure that the expected rewards with different numbers of neurons are increased. However, the small or large number of neurons does not regularly affect the expected rewards. Hence, under these conditions, the convergence time is different. From the figure, we can see that the curve with 8 × 4 and 4 × 2 neurons fluctuate within 40,000 to 60,000 steps. This means when the number of neurons is too small, the extracted feature information is not sufficient. Since the curve with 8 × 32 and 32 × 64 neurons has the best performance, we will adopt the number of neurons in the first and second graph convolutional layer (=8 × 32, 32 × 64) in the following experiments. Although the curve with 8 × 64, 64 × 128 consumes less time to convergence, the maximum of expected reward is less than the curve with 8 × 32, 32 × 64.

Furthermore, we performed an experimental verification of the convergence performance in case of the changed user distance distribution. The movement trajectories of all PUs and SUs within 1000 steps of the learning process are shown in Figure 11. We sampled the users’ specific locations every 50 steps. Hence, each user’s 20 changes of position are illustrated in the figure. The region of [0: 500, 0: 1000] represents the movement area of 4 PUs. The region of [500: 1000, 0:500] is the covered area of SBS1, and the region of [500: 1000, 500: 1000] is the covered area of SBS2. We simulated all the users’ movements in the way of pedestrians, following the random walk model. Moreover, we defined a limitation that SUs and PUs do not move beyond the boundaries of their respective cells. If there was a transboundary action, we discarded the action until there was a reasonable movement.

Figure 12 shows the convergence performance of different numbers of neurons in the case of the changed user distance distribution. The expected reward per training with increasing training iterations is depicted. It can be seen that there is the same trend in Figure 12. From the figure, the cumulative rewards increase as training continues, despite some fluctuations due to the users’ mobility. The underlay CRN is highly dynamic, including the channel state and network topology, which causes a large state space. Moreover, the underlying environment of each step is not exactly the same, which leads to the nuance of the expected rewards. In addition, the achievable data rate of SUs and overall system are compared in Figure 13.

## 5. Conclusions

In this paper, we proposed a channel selection and power adaptation scheme for the underlay CRN, maximizing the data rate of all SUs and guaranteeing the QoS of PUs. We adopted the DRL framework to explore the optimal resource allocation strategy. In this framework, the environment of the undelay CRN is the model as dynamic graphs, and the random walk model is used to imitate the users’ movements. Moreover, the crucial interference features of the constructed dynamic graph are extracted by the GCN. Further, an end-to-end learning model was designed to implement the following resource allocation task to avoid the split with mismatched features and tasks. The simulation results verified the theoretical analysis and prove that the proposed algorithm has stable convergence performance. The experiments show that the proposed algorithm can significantly optimize the data rate of CR networks and ensure the QoS requirements of PUs.

## Figures and Tables

**Figure 1 sensors-20-05216-f001:**
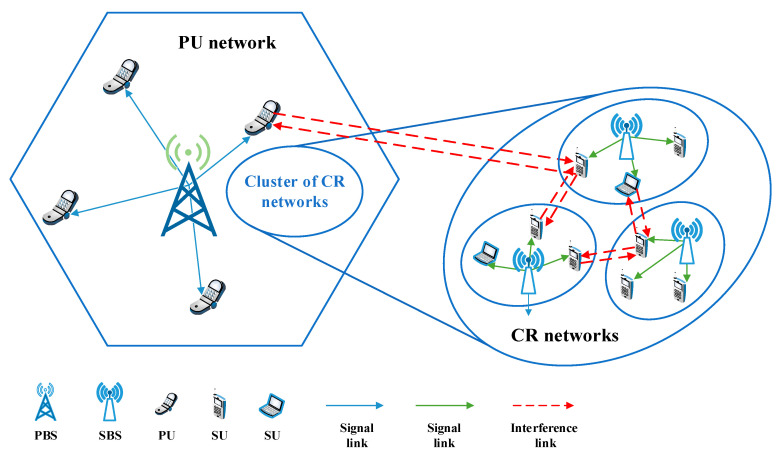
System model.

**Figure 2 sensors-20-05216-f002:**
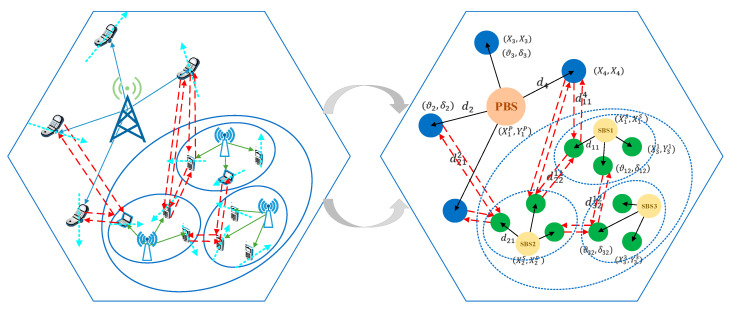
The method of dynamic graph construction.

**Figure 3 sensors-20-05216-f003:**
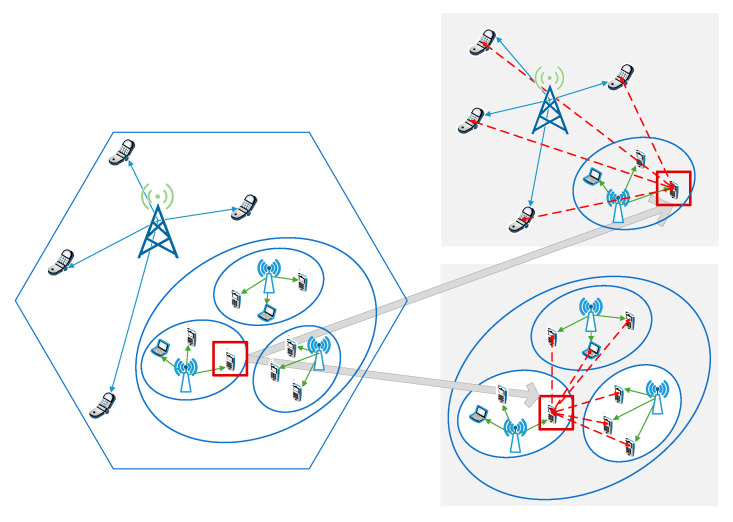
Analysis of co-channel interference that one secondary user may suffer.

**Figure 4 sensors-20-05216-f004:**
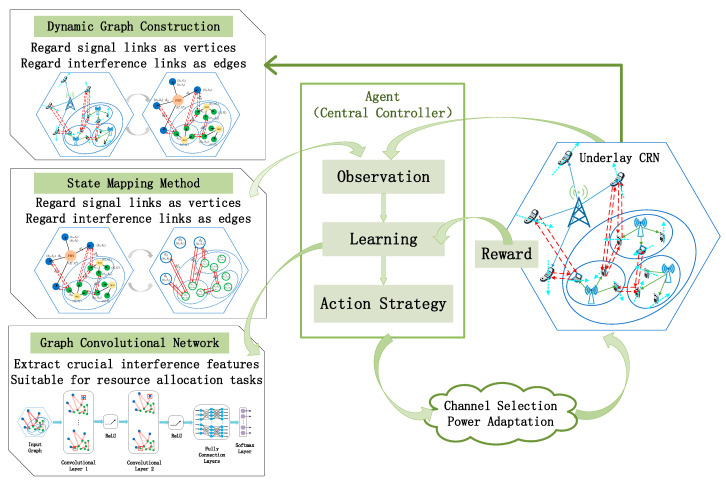
The deep reinforcement learning framework of the resource allocation for the underlay cognitive radio network.

**Figure 5 sensors-20-05216-f005:**
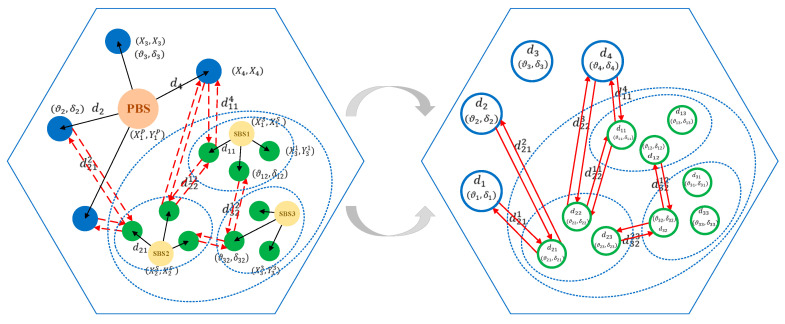
The state mapping method based on a dynamic graph.

**Figure 6 sensors-20-05216-f006:**
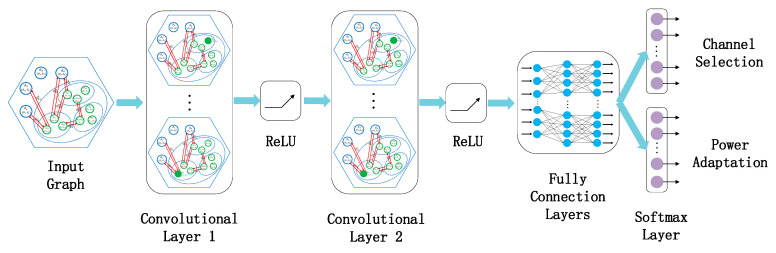
The network structure of the end-to-end learning model.

**Figure 7 sensors-20-05216-f007:**
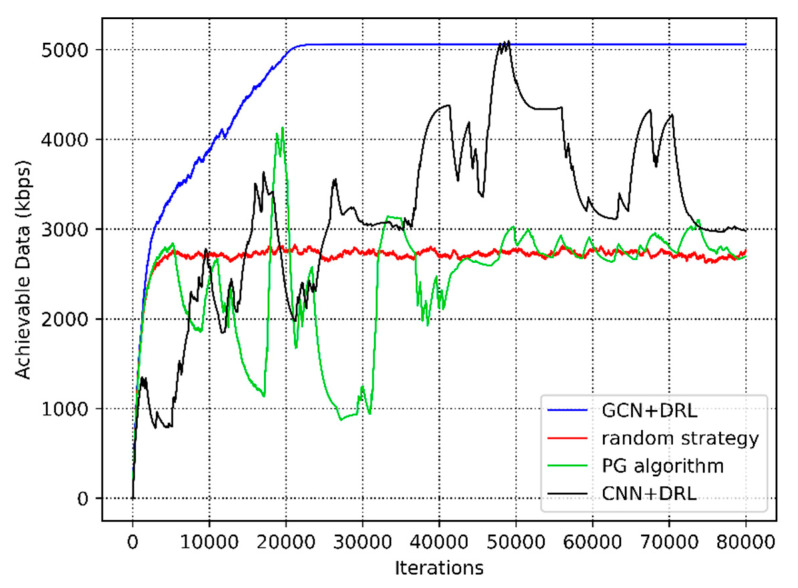
The achievable data rate of different algorithms.

**Figure 8 sensors-20-05216-f008:**
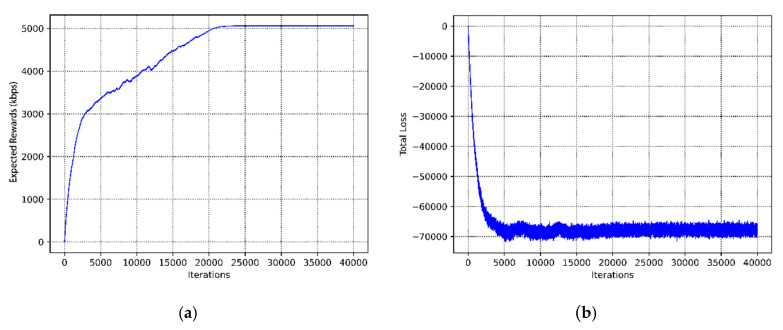
The convergence performance and training loss of our proposed algorithm. (**a**) The convergence performance of the resource allocation algorithm. (**b**) The training loss of the end-to-end learning model.

**Figure 9 sensors-20-05216-f009:**
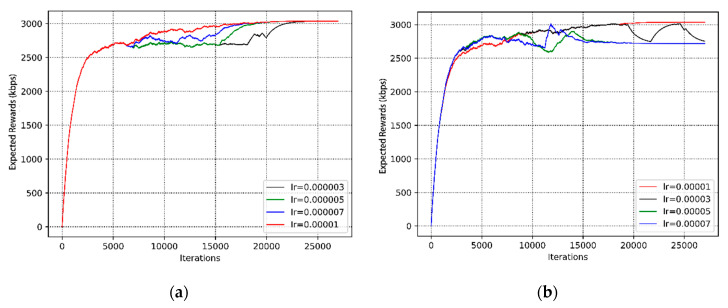
The convergence performance of different learning rates. (**a**) The convergence performance of the first set of different learning rates. (**b**) The convergence performance of the second set of different learning rates.

**Figure 10 sensors-20-05216-f010:**
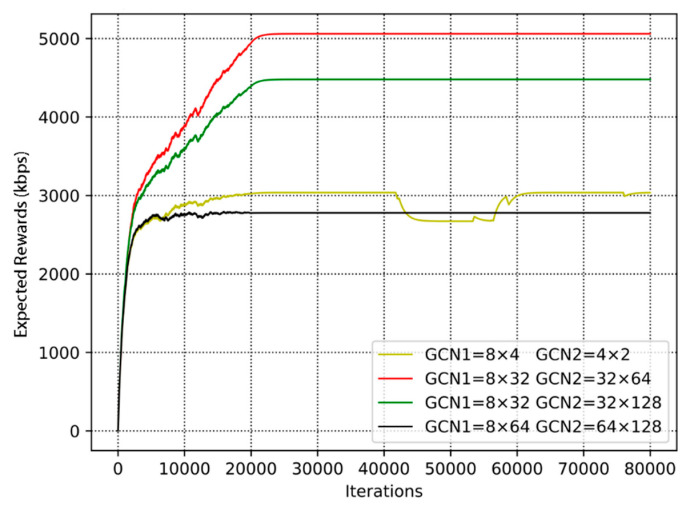
The convergence performance of different numbers of neurons.

**Figure 11 sensors-20-05216-f011:**
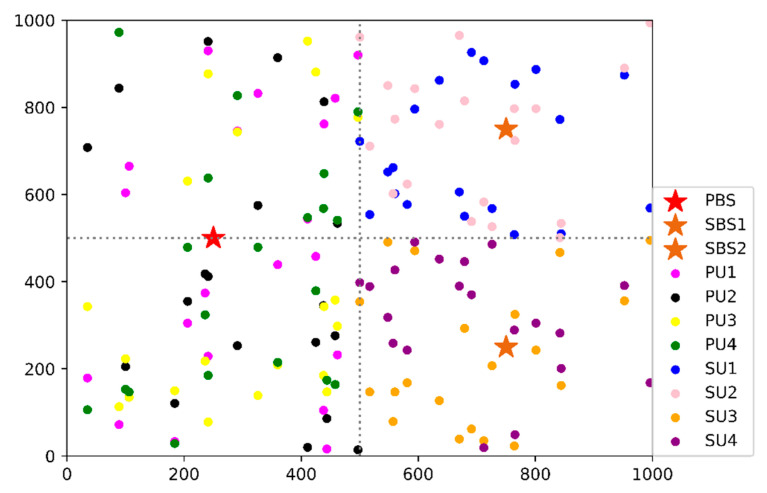
The movement trajectory of primary users and secondary users within 1000 steps.

**Figure 12 sensors-20-05216-f012:**
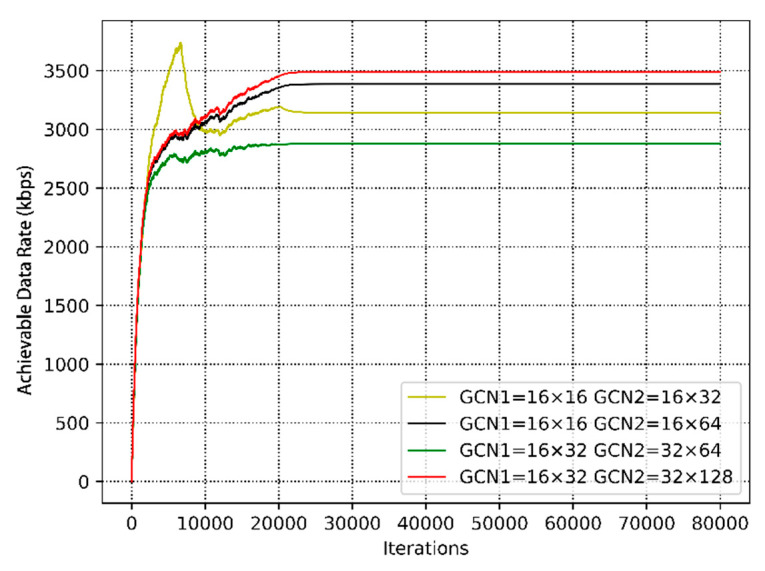
The convergence performance under users’ movements.

**Figure 13 sensors-20-05216-f013:**
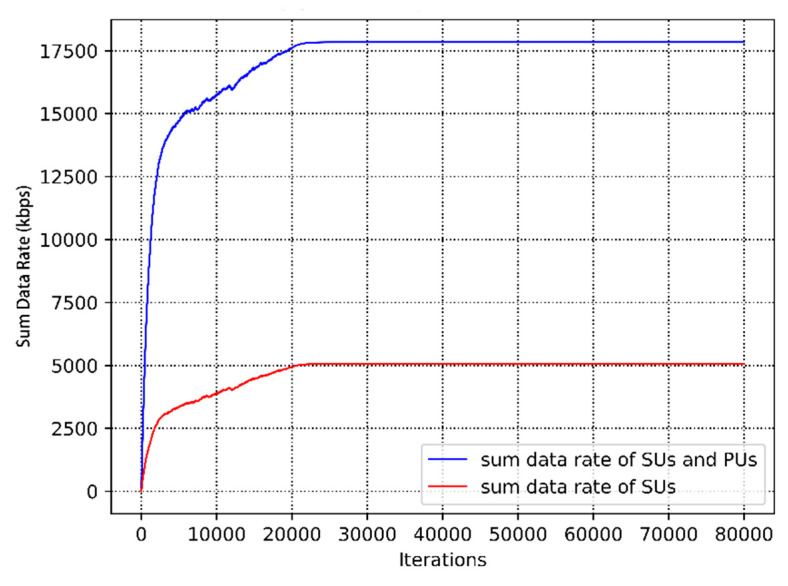
The achievable data rate of the underlay cognitive radio network.

**Table 1 sensors-20-05216-t001:** Simulation parameters.

Parameter	Value
Cell radius	500 m
BS antenna gain	18 dBi
User antenna gain	3 dBi
Carrier frequency	2 GHz
Path loss model	137.3 + 35.2 log(d(km)) (dB)
Noise power	−122 dBm
Interference temperature	6 db
RB bandwidth	180 kHz
Number of RBs	8
Transmission power	[3,13,23] dBm
Number of PUs	4
Number of CR network	2
Number of SUs per CR network	2
Direction of user movement	[0, 2π]
Speed of user movement	4.3 km/h
Discount factor	0.995

**Table 2 sensors-20-05216-t002:** A comparison of different algorithms for resource allocation in the underlay cognitive radio network.

	Random Strategy	PG Algorithm	CNN + DRL	GCN + DRL
Neural networks used	GCN	MLP	CNN	GCN
DRL framework	🗴	🗸	🗸	🗸
Computational complexity	$O\left({\left(NM\right)}^{\left(U+SV\right)}\right)$	$O\left({\left(NM\right)}^{\left(U+SV\right)}\right)$	$O\left({\left(NM\right)}^{\left(U+SV\right)}\right)$	$O\left({\left(NM\right)}^{\left(U+SV\right)}\right)$
Convergence time	1590 s	17,000 s	≥30,880 s	8800 s
Optimal solution	🗴	🗴	🗴	🗸
Scalability	🗴	🗴	🗴	🗸

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
