# Peer review of "A Graph Convolutional Network-Based Deep Reinforcement Learning Approach for Resource Allocation in a Cognitive Radio Network"

_sensors, 2020, doi:10.3390/s20185216_

Round 1
Reviewer 1 Report
Authors propose DRL-GCN model for resource allocation in cognitive radio network which can be a solution for the deployment of 5G. This paper address the previous and related works well, and also description for system model and problem formulation is very informative. If the following issues are clarified, this paper is worth reading for communication engineers:
1) It is necessary to analyze the simulation results and to find the meaning of them. For example, analysis about Fig. 7 deals with the convergence time according to methods. I think that it is also meaningful to compare the computational complexity or time rather than iteration numbers.
2) Authors compare the convergence performance according to learning rates in Fig. 9. I wonder how can find the (sub)optimal learning rate.
3) It is meaningful to simulate the total loss function addressed in lines 461~463.
4) Minor comments
- Line 323 : 4th SU (covered by sth SBS)
- Units : y-axis in Figs. 7~10 and 12~13
- Consistent terminology
* DRL+GNN (line 434) vs. GNN+DRL(lines 472, 475), legend in Fig. 7
* GCN vs. GNN : GNN includes GCN. In this paper, GCN is preferred to GNN according to title of paper.
- The title above each figure(Figs. 7~13) is useless because the same content is addressed in the caption.
- Notation : Set excluding element ==> {S}\s not {S}/s (e.g., line 184, eqn. (2), (9), …)
5) Typo
- line 430 : carried0 -? Carried
Reviewer 2 Report
Well-structured and expounded paper with important innovations and findings. Graph convolutional neural networks which have very recently found multiple applications in wireless networks are trained using an effective reinforcement learning approach. In addition, the graph’s construction method is practical. The system model of underlay cognitive radio network is important for modern communication networks and the proposed algorithm yields adequate results for channel and power allocation.
Some suggestions and questions to improve the paper’s quality and clarity:
- I suggest proof-reading for very minute vocabulary issues throughout the paper.
- As for the interference temperature threshold (first introduced on lines 217-219), it would be valuable to detail in a few words how it is determined for the simulation.
- On lines 415 and 416 it is said that two convolutional layers are used for effectiveness. Is this determined empirically? Can it be said how using less or more layers will affect the performance? In addition, on lines 422 and 423 it is said the convoloutional and fully connected layers are shared. What is meant exactly by sharing them? In what way are they shared? Perhaps adding a small discussion on these points may be useful.
- On line 470 it is said that the simulation parameters are taken from reference [26]. I could not find the part of the book in which they are listed. I believe mentioning the chapter or part of this foundational book would be helpful in comparing the utilized system model to others, and for better clarity.
- In Table 2, the noise power and transmission powers are given in dB. Perhaps it is meant that they are in dBm? If it is in dBm, then if I am correct in my calculations, the noise power for bandwidth of 180 kHz would be -122 dBm. In addition, how is the discount factor determined? What is the center frequency?
- On line 478 and the ordinate axis of Fig. 7, 8, 9, 10, 12 and 13, what is the measurement unit of the Achievable Data Rate, Sum Data Rate and Expected Rewards – bps or perhaps an arbitrary relative unit? I believe it should be explicitly stated on at least one place for technical clarity.
- On lines 498 and 499, most of the sentence just reiterates the previous one, so it should better be rephrased.
- The learning rates for which the model is examined, are 0.00001, 0.00003, 0.00005, 0.00007. To the best of my knowledge the learning rates in variety of deep learning approaches are around 0.001 or higher. Perhaps a short clarification on why such learning rates are used can be added.
- In Fig. 11, it is seen that SUs and PUs do not move outside the boundaries of their respective cells. Is this a specifically-defined limitation on their movement or a result of the algorithm's operation? Do they reach outside their cells' boundaries after more than 1000 steps?
